# The Role of Vitamin K Deficiency in Chronic Kidney Disease—A Scoping Review

**DOI:** 10.3390/nu17152559

**Published:** 2025-08-05

**Authors:** Valdemar Tybjerg Wegge, Mette Kjær Torbensen, Allan Linneberg, Julie Aaberg Lauridsen

**Affiliations:** 1Faculty of Health and Medical Sciences, University of Copenhagen, 2200 Copenhagen, Denmark; pdx435@alumni.ku.dk (V.T.W.); pqk293@alumni.ku.dk (M.K.T.); 2Center for Clinical Research and Prevention, Copenhagen University Hospital, Bispebjerg and Frederiksberg, 2000 Copenhagen, Denmark; julie.aaberg.lauridsen@regionh.dk; 3Department of Clinical Medicine, Faculty of Health and Medical Sciences, University of Copenhagen, 2200 Copenhagen, Denmark

**Keywords:** vitamin K, chronic kidney disease, supplementation

## Abstract

**Background/objectives**: Chronic kidney disease (CKD) affects up to 15% of the global population and is driven by vascular and interstitial damage, and is most prevalent in persons with hypertension and diabetes. Vitamin K, a necessary cofactor for activation of vitamin K-dependent proteins may modulate these processes. It is well established that vitamin K deficiency is associated with CKD, but the therapeutic effects of supplementation on kidney function are still uncertain. We aimed to review the current evidence on the effect of vitamin K deficiency and supplementation on any marker of renal function and kidney disease, across general adult populations and CKD patient populations. **Methods**: A search was conducted in PubMed, targeting terms related to vitamin K status and CKD. Studies were included if they reported data on vitamin K status or supplementation in relation to kidney function outcomes. **Results**: A total of 16 studies were included. Nine interventional studies were included and confirmed that vitamin K supplementation improves biomarkers of vitamin K status but showed no consistent beneficial effects on renal function. Seven observational studies across populations found significant associations between vitamin K status and decline in kidney function; however, associations were often attenuated after adjustments. **Conclusions**: No clear effect of supplementation was observed on the reported kidney markers in patient populations. A clear association between low vitamin K status and impaired kidney function was confirmed. Studying heterogeneity makes the comparability and generalizability of the results difficult. Our review highlights the need for more cohort studies and clinical trials in general or patient populations.

## 1. Introduction

Chronic kidney disease (CKD) is estimated to have a global prevalence of up to 15%, making it a major public health burden [1]. CKD is a collective term for several diseases causing progressive decline of the glomerular filtration rate (GFR). There are multiple risk factors for developing CKD, with the most prevalent being vascular and interstitial damage, particularly caused by diabetes and hypertension [2].

Research has already established a high prevalence of vitamin K deficiency in CKD patients. This is believed to be partly due to dietary restrictions, medication-induced, and possibly impaired vitamin K recycling in CKD patients [3,4]

Some of the early vascular changes in hypertension leading to damage in the kidneys are the thickening of the tunica media and a reduced luminal diameter causing increased vascular resistance. The vascular modeling is particularly present in the small resistance arteries, such as the proximal arteries of the kidneys [5]. The increased resistance results in shear stress on the endothelium, promoting endothelial dysfunction and upregulation of pro-inflammatory cytokines leading to the activation of several transcription factors, including nuclear factor kappa beta (NF-kB) and bone morphogenic protein 2 (BMP2). This initiates a phenotype switching in the vascular smooth muscle cells (VSMC) from a contractile to a pro-inflammatory and osteogenic phenotype. The differentiation of VSMCs increases calcification and along with increased vascular shear stress mediating kidney damage [6,7].

Diabetes is another prominent risk factor for CKD. Approximately 40% of individuals with type 2 diabetes (T2D) and 30% with type 1 diabetes (T1D) develop diabetic kidney disease [8]. T2D is a chronic metabolic condition characterized by insulin resistance and hyperglycemia, and T1D is an autoimmune disease attacking the insulin-producing beta cells of the pancreas [9,10]. The pathophysiology behind diabetic nephropathy includes renal structure changes leading to glomerular hyperfiltration, albuminuria, and decline in GFR. In addition to these structural changes, metabolic alterations result in glomerular hypertrophy, inflammation of the renal interstitium, and fibrosis [8].

Vitamin K is a group of fat-soluble molecules with phylloquinone (vitamin K1) particularly present in green leafy vegetables and the menaquinones (MKs), also referred to as vitamin K2, present in fermented foods and animal-based products including dairy products [11]. Vitamin K2 consists of subtypes MK4-MK13 and both vitamin K1 and vitamin K2 exert their effects through gamma-glutamate carboxylase, activating vitamin K-dependent proteins (VKDPs) [10]. Data on MK vitamers in food is limited and a lack of international consensus on reporting standards complicates accurate intake assessment [11]. MKs generally have longer half-lives than vitamin K1. MK-7 has a half-life of 3 days, while K1 is cleared within 1–2 h. As fat-soluble compounds, MK absorption depends on dietary fat and varies across vitamers. Compared to K1, MKs are typically considered more bioavailable for the body to absorb, as K1 is tightly bound in plant matrices, though its absorption can improve significantly when consumed with fat [10].

There is no generally accepted standard for how to measure vitamin K status. However, measurement of blood levels of the inactive (uncarboxylated) VKDPs, dephospho-uncarboxylated matrix Gla protein (dp-ucMGP), and protein induced by vitamin K absence (PIVKA), are thought to provide a better measure of vitamin K status compared to direct measurement of circulating vitamin K vitamers, which have a short half-life in blood. Several studies have found a strong decreasing effect of vitamin K supplementation on dp-ucMGP levels in a dose-dependent manner, as a reflection of improved vitamin K status [12,13].

In particular, the two VKDPs, MGP and growth arrest-specific gene 6 (GAS-6) are believed to play a role in the pathogenesis of CKD. GAS-6 is expressed in the endothelium, VSMC, and bone marrow. It has a wide range of effects on hemostasis and inflammation, suppressing the pro-inflammatory cytokine NF-kB [14]. MGP is expressed by VSMC and chondrocytes [15]. It is a potent inhibitor of vascular calcification by binding BMP-2 thus protecting against VSMC differentiation. Furthermore, MGP’s chemical structure binds calcium crystals preventing calcium depositing in the vascular tissue [12].

Studies have found associations between low vitamin K status and CKD, as plasma dp-ucMGP levels have been found to increase with the stage of the disease [15]. This is believed to be due to decreased vitamin K intake due to a potassium-restrictive diet, adverse effects of medications, and disease-related impaired vitamin K metabolism [16]. A review article from 2021 highlights CKD patients as having an increased risk of arterial calcification, potentially caused by vitamin K deficiency, underlining the interest in investigating vitamin K supplementation as a therapeutic intervention in CKD patients [17]. One scoping review highlighted the potential of vitamin K as a therapeutic target in CKD, though optimal dosing remains unclear [18]. Supplementation in hemodialysis patients reduced dp-ucMGP levels and slowed vascular calcification progression, especially in those with existing calcifications [19]. However, a review from 2021 concluded that there is no strong evidence that vitamin K supplementation slows progression of calcification in CKD patients, but that the supplementation is safe and improves serum markers [17].

Vitamin K also shows potential as a nephroprotective measure by regulating the tissue-damaging pathogeneses. As MGP is a potent inhibitor of vascular calcification, decreased levels of the vitamin K-dependent protein can increase the medial artery calcification in the renal arteries, potentially impairing the renal perfusion and the renal function [20]. A study from 2022 found MGP-deficient mice to have an increased collagen-producing myofibroblasts and tissue fibrosis in the kidneys, pointing to a protective role of vitamin K-dependent proteins in the progression of CDK [21].

Few studies have also investigated the effect of vitamin K on glycemic status, inflammation, and insulin resistance. A study from 2018 found that supplementation with vitamin K1 reduced the activation of the NF-kB pathway and secretion of pro-inflammatory cytokines. Furthermore, they found that, within a patient population with T2D, there was a strong association between low levels of vitamin K1 and increased insulin resistance. Vitamin K1 supplementation was found to reduce insulin resistance and enhance glycemic status in both mice and patients with T2D, supporting the potentially preventive properties of vitamin K in subjects with T2D [22]. A review from 2016 found that supplementation with vitamin K1 reduced the expression of the pro-inflammatory cytokine IL-6, contributing to the potential anti-inflammatory properties of vitamin K [23]. Figure 1 highlights the potential key mechanistic pathways as to how vitamin K can affect the progression of CKD.

A recent cross-sectional general population study in adults found that increasing levels of plasma dp-ucMGP were linked to central obesity, diabetes, hyperlipidemia, and reduced kidney function [24]. Thus, one doubling of dp-ucMGP was associated with a tenfold (OR 9.83, 95% CI 5.49–17.59) increased risk of reduced kidney function. Interestingly, within the subgroup of hypertensive participants, the association between lower vitamin K status and increased risk of reduced kidney function was even higher [24]. In conclusion, these studies suggest a potential role of vitamin K in kidney disease. However, the exact mechanisms and clinical importance of vitamin K in kidney disease are still not clear. While CKD and impaired kidney function seem to be associated with vitamin K deficiency, less is known about whether vitamin K deficiency is associated with progression of CKD and whether improvement of vitamin K status, e.g., by supplementation, can prevent or change the course of CKD.

We aimed to perform a scoping review investigating the current evidence on the effect of vitamin K deficiency and supplementation on renal function across general adult populations and the CKD patient populations.

## 2. Materials and Methods

This scoping review was conducted in agreement with Joanna Briggs Institute’s methodology using the PRISMA extension for scoping reviews (PRISMA-ScR) [25]. A protocol including an initial limited search was conducted prior to the final literature search. The final search strategy was based on biomarkers and terms relating to CKD and renal function in one mesh term category and biomarkers and terms relating to vitamin K in the second (Appendix A, Table A1 and Table A2).

We utilized PubMed database to identify studies that investigated the effect of vitamin K supplementation or vitamin K deficiency on markers of kidney function, progression of kidney disease, or kidney transplantation. We included studies on general adult populations, with normal kidney function, and CKD patient population, with or without other known diseases. The search included cohort studies or clinical trials, published in English regardless of date of publication or geographical location.

All identified articles/citations were uploaded into Covidence, and duplicates were removed [26]. Titles and abstracts were independently screened by two reviewers, M.K. Torbensen and V.T. Wegge, and selected articles were full-text screened. Disagreements were resolved through discussion. Final articles for inclusion were additionally examined by an experienced third reviewer, J.A. Lauridsen (Figure 2). We used and modified a Covidence data extraction tool. The extracted data included information on participants, study design, and key findings relevant to this scoping review’s objective.

The reference list of all included sources of evidence was screened for potential additional studies by screening cross-references.

## 3. Results

Our literature search identified 2126 studies from the PubMed database of which three were duplicates. Through title and abstract screening, 2064 articles were deemed irrelevant to the present review, leaving 59 articles for full-text screening. Following full-text screening, a total of 39 articles were excluded, 13 due to non-relevant outcomes, three studies due to wrong interventions, and 23 due to non-relevant study designs. We cross-referenced the 59 articles and found one article not yet included in our search. The third reviewer screened the final 20 articles which led to the exclusion of five articles due to non-relevant study designs. Thus 15 articles were included from the literature search and one from cross-referencing, leaving 16 articles for final extraction (Figure 2). The 16 articles consisted of nine clinical trials (Appendix B, Table A3) and seven cohort studies (Appendix B, Table A4). A focused comparative table of the clinical trials is provided to highlight the heterogeneity of the study designs (Appendix B, Table A5). The clinical trials consisted of eight randomized controlled trials (RCTs) and one non-controlled trial in patients. Three studies were conducted in populations of kidney transplant recipients (KTRs), three in patients with varying degrees of chronic kidney disease (CKD 3-5), and three in hemodialysis (HD) patients. The cohorts consisted of three studies in adult general populations, and four in patient populations (KTR (two studies), T1D (one study), and CKD patients (one study)). Nine of the 16 included studies had one or more primary outcomes relating to kidney function (transplant failure (two studies), decline in kidney function, and incident CKD or progression of CKD (five studies)). Thirteen studies had dp-ucMGP, ucMGP, or MGP levels, as measurements for vitamin K status. Four studies had uncarboxylated osteocalcin (ucOC) and/or the ucOC/OC ratio as markers, one study had phylloquinone levels as a marker, and finally two studies had no measurement for vitamin K status. Two studies had a primary outcome relating to the associations of dp-ucMGP levels and kidney function (one study) or dp-ucMGP levels and dietary intake of vitamin K1 and vitamin K2 (one study). The remaining studies included secondary outcomes relating to kidney function.

### 3.1. Experimental Studies

#### 3.1.1. Chronic Kidney Disease

Three RCTs in CKD patients investigated the effects of vitamin K supplementation on the progression of kidney disease. Witham et al. performed an RCT that found no treatment effect on the estimated glomerular filtration rate (eGFR) or urine albumin–creatinine ratio (UACR) after 400 µg/day MK-7 supplementation for 12 months in 159 adult patients with CKD (*n* = 80 intervention, *n* = 79 placebo) [27]. In two separate RCTs, Kurnatowska et al. found that dp-ucMGP levels declined significantly following supplementation with 90 µg/day MK-7 + 10 µg vitamin D for 270 ± 12 days compared to the control group receiving only vitamin D supplementation. They included 42 stage 3–5 patients in 2015 (*n* = 29 intervention, *n* = 13 placebo) and 38 stage 4–5 patients in 2016 (*n* = 26 intervention, *n* = 12 placebo). The trial from 2015 found a significant between-groups difference in eGFR after intervention, with a decline in eGFR in the intervention group; although this group had a lower eGFR at baseline when compared to the placebo group, they did not adjust for the baseline difference. They also found a significant rise in creatinine in the intervention group which was borderline significant in the between-groups difference. The trial in 2016 showed a strong inverse association between eGFR and dp-ucMGP levels, and found correlations between dp-ucMGP levels, creatinine, and proteinuria, but no significant between-group change in eGFR between the supplementation and placebo groups [28,29].

#### 3.1.2. Kidney Transplant Recipients

Lees et al. performed an RCT examining the effect of 5 mg menadiol diphosphate (a vitamin K analog) thrice weekly for 12 months on the kidney outcomes proteinuria, eGFR, and urine protein–creatinine ratio in 90 KTR (*n* = 45 intervention, *n* = 45 placebo). They observed no treatment effect on eGFR or proteinuria [30]. Eelderink et al. also reported no significant change in kidney function (eGFR or creatinine clearance) in their RCT following 360 µg/day MK-7 treatment for 12 weeks in 40 patients (*n* = 20 intervention *n* = 20 placebo) [31]. Mansour et al. performed a non-controlled clinical trial supplementing 60 KTR patients (no control group) with 360 µg/day MK-7 for 8 weeks. They found a small, but statistically significant, increase in serum creatinine, indicating a decline in kidney function, although no adjustments were made for this outcome [32].

#### 3.1.3. Hemodialysis Patients

Three RCTs in HD populations assessed serum creatinine after different supplementation strategies. Naiyarakseree et al. performed an intervention with supplementation of 375 µg/day MK-7 for 24 weeks in 96 patients (50 intervention, *n* = 46 control group), Oikonomaki et al. supplemented 200 µg/day MK-7 for 12 months in 102 patients (*n* = 58 intervention, *n* = 44 control group), and Macias-Cervantes supplemented 10 mg intravenous vitamin K1 thrice weekly for 12 months in 60 patients (*n* = 30 intervention, *n* = 30 placebo). All three studies had creatinine as the only marker of kidney function, and none of the studies showed a significant between-groups difference following intervention [33,34,35].

### 3.2. Cohort Studies

#### 3.2.1. General Adult Populations

Three cohort studies in the general adult population examined associations between vitamin K status and kidney function. Groothof et al. assessed a cohort of 3969 adults with a mean follow-up period of 7.1 years [36]. In the crude analyses, Groothof et al. found strong associations between high dp-ucMGP levels at baseline and incident CKD and microalbuminuria, but these associations disappeared when adjusting for baseline eGFR [36]. O’Seaghdha et al. assessed 1442 adults with a mean follow-up of 7.8 years. They found a higher risk of CKD and microalbuminuria with higher vitamin K1 levels (four quartiles of phylloquinone levels) at follow-up. These associations remained significant after multivariable adjustment including eGFR [37]. Wei et al. assessed 1009 adults with a median follow-up of 8.9 years. They identified an inverse association between dp-ucMGP levels and eGFR, with high baseline dp-ucMGP levels (indicating low vitamin K status) predicting eGFR decline, though adjustment for baseline eGFR was not specified [38].

#### 3.2.2. Cohort Studies in Kidney Transplant Recipients

Two cohort studies examined kidney transplant recipients. Van Ballegoijen et al. assessed 461 KTRs with stable kidney function with a median follow-up period of 9.8 years. They found that elevated dp-ucMGP levels (dp-ucMGP > 1057 pmol/L) in combination with both low and high vitamin D levels were associated with higher hazard ratios for death-censored graft failure (defined as return to dialysis therapy or re-transplantation), compared to the group with lower dp-ucMGP levels (dp-ucMGP < 1057 pmol/L) both in crude and when adjusting for age, sex, any cyclic variation, current smoking status, body mass index, triglycerides, average blood glucose levels, systolic blood pressure, year of transplantation, dialysis duration, and eGFR [39]. Keyzer et al. assessed 518 KTRs for a mean period of 9.8 years. They reported an association between higher quartiles of dp-ucMGP and transplant failure (defined by return to dialysis therapy or re-transplantation), though the association reduced with adjustment models and finally became non-significant after adjusting for baseline eGFR [40].

#### 3.2.3. Cohort Study in Type 1 Diabetes Patients

In a cohort of 638 patients with T1D followed for 5–7 years, Nielsen et al. found that higher quartiles of dp-ucMGP (low vitamin K status) were associated with a markedly increased risk of incident end-stage kidney disease (ESKD). This association remained in crude and partially adjusted models but was lost after adjusting for baseline eGFR, albuminuria, and T1D duration [41].

#### 3.2.4. Cohort Study in Chronic Kidney Disease Cohort

Roumeliotis et al. followed 66 type 2 diabetic CKD patients for 7 years, reporting positive correlation between dp-ucMGP levels and proteinuria. They found an inverse correlation between dp-ucMGP levels and both baseline and follow-up eGFR. Higher dp-ucMGP levels (≥656 pmol/L) were associated with a 4.02 times higher longitudinal risk of a ≥30% reduction in eGFR or progression to ESKD in a multivariate model adjusted for T2D duration, serum albumin, and proteinuria [42].

## 4. Discussion

Vitamin K supplementation in eight RCTs and one non-controlled trial consistently reduced dp-ucMGP levels in CKD patients, KTRs, and HD patients, but no beneficial changes in eGFR, creatinine, or proteinuria were observed. Two studies showed a significant decline in kidney function after intervention. In general adult populations, three cohort studies reported associations between low vitamin K status, measured as high dp-ucMGP levels, and indicators of impaired kidney function in terms of reduced eGFR, an increased risk of incident CKD, and microalbuminuria. While some of these associations attenuated after adjusting for baseline kidney function, others remained significant even after multivariable adjustment [36,37,38]. In kidney transplant recipients, one study reported a significant association between high dp-ucMGP levels (low vitamin K status) and graft failure after full adjustment, while others found no difference or attenuated effects after adjustment. Among patients with T1D or CKD, higher dp-ucMGP levels were associated with an increased risk of ESKD and eGFR decline. The associations attenuated in the T1D group after adjustments and whilst the CKD group’s association remained significant, they reported a very limited adjustment model.

The nine experimental studies differed with regard to type, administration, dose, and duration of vitamin K supplementation. Across the studies they used vitamin K1, MK-7, or menadiol diphosphate either as oral tablets or intravenous administration. The dosages varied from 90 µg to 400 µg daily with a duration of intervention ranging from 8 weeks to 12 months. The heterogeneity of the studies limits the comparability and generalizability of the results. Only one of the studies were designed to investigate progression of CKD and progression of kidney function as primary outcomes, though with the lowest dose of supplementation. Of the eight included RCTs and one non-controlled clinical trial, six out of nine studies employed plasma dp-ucMGP as a marker of vitamin K status. These six studies all demonstrated significant lowering of dp-ucMGP levels in the intervention groups, demonstrating a biological effect. However, no studies reported any significant changes or tendencies of improvements in renal endpoints such as eGFR, proteinuria, or creatinine, and two studies found a significant increase in creatinine after intervention, indicating a worsening of kidney function. This could suggest that supplementation of vitamin K does not prevent progression of kidney disease. Two studies even indicated a decrease in kidney function, but both studies had limitations; one did not include a control group and the other appeared to have marked differences in baseline characteristics between the intervention and control group. Arguably, this also points to gaps in the literature and the need for more studies; e.g., the dose and duration employed in the studies may have been insufficient to influence renal outcomes beneficially. CKD patients are known to be vitamin K-deficient and could potentially require higher doses of vitamin K supplementation to sufficiently cover the biological functions requiring vitamin K. Furthermore, due to the renal impairment and thus poor excretion of phosphate, many CKD patients are treated with a phosphate binder, inhibiting the absorption of phosphate in the digestive system. Phosphate binders are known to also cause impaired absorption of vitamin K, thus potentially confounding the effect of supplementation [43]. The results could mean that the beneficial effects of vitamin K supplementation are limited in these patient populations due to their pre-existing kidney disease and impaired vitamin K function. Several other factors likely also play a role in the development of functional vitamin K deficiency. A study by Kaesler et al. found that the pharmacokinetics of vitamin K (K1, MK-4, and MK-7) are altered in patients with uremia, a known complication of CKD. In their prospective clinical trial, they observed changes in high-density lipoprotein (cholesterol) particles in patients with uremia causing reduced uptake of MK-7, impairment of the vascular protective function, and potentially contributing to calcification of the arteries. Furthermore, in the same study, clinical trials in CKD rats showed altered vitamin K recycling, an essential part of the vitamin K cycle [16]. These findings could indicate that CKD patients may not benefit from vitamin K supplementation due to decreased bioavailability and reuse of vitamin K. Whether these barriers may be overcome by increasing the dose of vitamin K in interventional studies is not known and should be further studied in interventional studies. However, as CKD is often a slowly progressing disease [44], investigation of healthy subjects or patients in early stages of CKD is difficult in clinical trials. This makes cohort studies in a general population more feasible for this research question. Another possibility for examining the potential preventive effects of vitamin K supplementation on CKD would be to include patients with diabetes or hypertension, as the incidence of CKD is higher within these groups. An experimental study in rats from 2015 showed a nephroprotective effect of vitamin K1 supplementation after streptozotocin-induced destruction of the beta cells in the pancreas. Supplementation had significant beneficial effects on insulin levels, blood glucose, creatinine, the albumin–creatinine ratio, urea, and uric acid. It also showed a neutralization effect on histopathological change, and immunohistochemical changes in the kidneys of the vitamin K-supplemented rats. The pro-inflammatory NF-kB was expressed widely in the streptozotocin-induced rats and completely absent in the supplemented rats [45]. These results could point towards vitamin K being a protective factor against the adverse effects of diabetes on kidney function.

The general adult population studies reported associations between low vitamin K status and several kidney disease outcomes; however, in several of the studies, these associations became non-significant after adjustment for baseline eGFR. The fact that most studies lost significance after adjusting for baseline eGFR could either point to the baseline kidney function driving the effect of vitamin K on the progression of kidney disease or potentially mediating the effect. In case of the latter, adjusting increases the risk of missing a potential therapeutic effect of vitamin K supplementation. In general, the studies were not consistent in their approach when accounting for potential confounders. This could point to the interplay between vitamin K and kidney function being complex and further analyses being required to uncover the effects.

## 5. Conclusions

In this scoping review we investigated the role of vitamin K in the decline in kidney function and progression of CKD. Observational studies provided evidence of a temporal association between low vitamin K status and a decrease in kidney function.

Interventional studies did not suggest that vitamin K supplementation could prevent decline in kidney function markers. The populations and study settings in the interventional studies differed on several parameters including sample size and type, administration, dose, and duration of vitamin K supplementation. Study heterogeneity in the interventional studies makes the comparability and generalizability of the results difficult.

Our results indicate the interplay between vitamin K and kidney function is complex and in need of further studies to uncover whether vitamin K supplementation could prevent a decline of kidney function in patients or even prevent the development of CKD in high-risk groups. Both observational and interventional studies in larger patient groups with renal endpoints are needed to clarify the preventive and therapeutic potential of vitamin K supplementation.

## Figures and Tables

**Figure 1 nutrients-17-02559-f001:**
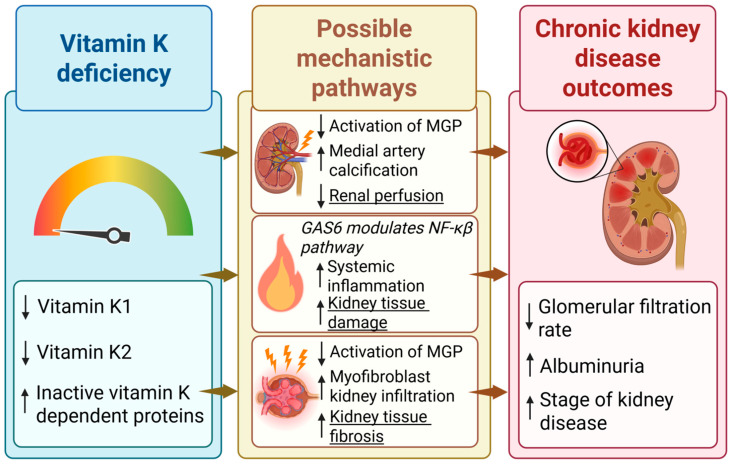
The proposed biological pathways through which reduced vitamin K status may contribute to the development and progression of chronic kidney disease (CKD). Low levels of vitamin K, either assessed as low vitamin K1 or vitamin K2 lead to reduced activation of vitamin K-dependent proteins such as matrix Gla protein (MGP) and growth arrest-specific protein 6 (GAS6), resulting in increased medial artery calcification [20], inflammation [22], and kidney tissue fibrosis [21]. These processes may impair renal perfusion and promote structural kidney damage, ultimately contributing to a decline in kidney function. ↑—increase; ↓—decrease.

**Figure 2 nutrients-17-02559-f002:**
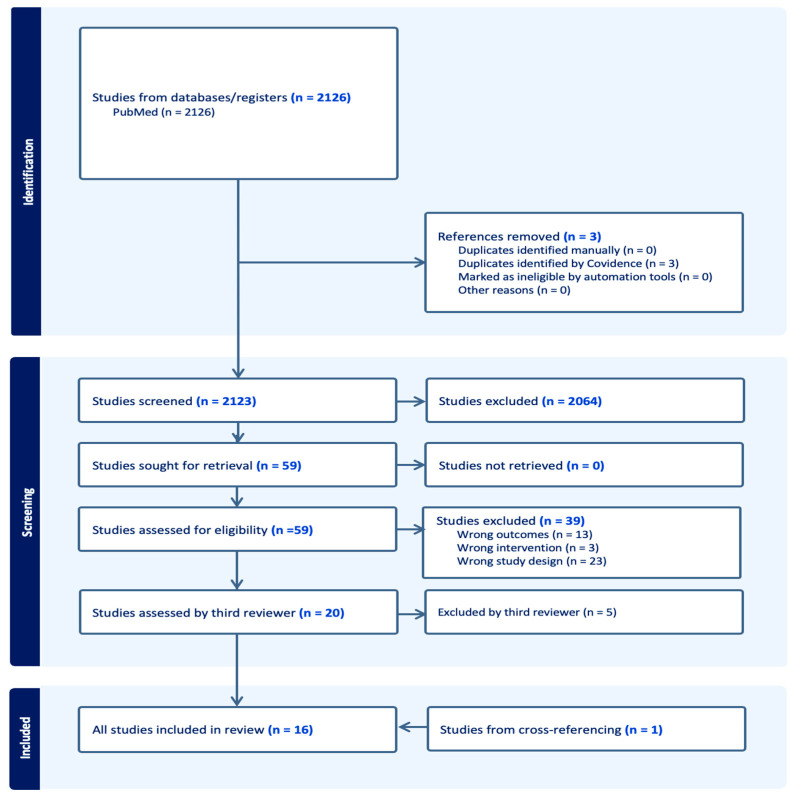
PRISMA flowchart of included/excluded studies.

## Data Availability

No new data was created or analyzed in this study. Data sharing is not applicable to this article.

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
