# Peer review of "The Role of Vitamin K Deficiency in Chronic Kidney Disease—A Scoping Review"

_nutrients, 2025, doi:10.3390/nu17152559_

Round 1
Reviewer 1 Report
Comments and Suggestions for Authors
The article is a thoroughly researched review that addresses the important and timely topic of the relationship between vitamin K deficiency and chronic kidney disease (CKD). The authors have attempted to evaluate both observational and interventional studies in relation to the potential effects of vitamin K supplementation on kidney function. Although the topic is interesting and worth exploring, the paper has several significant limitations that should be considered before publication.
Timeliness and relevance of the research problem:
The growing interest in the role of vitamin K in preventing the progression of CKD makes the topic valuable from a public health perspective.
Reliable review methodology:
The work was prepared in accordance with PRISMA-ScR recommendations and JBI guidelines, which adds to its credibility and transparency.
A wide range of studies were included – both observational and interventional, covering different populations (general, HD, KTR, T1D, T2D).
The authors do not jump to conclusions and point out the limitations of the available data and the need for further research.
Critical comments and recommendations:
Unclear main working hypothesis:
The title suggests an analysis of the direction of causality (vitamin K deficiency as ‘cause or effect’), but the content of the paper lacks a clear theoretical model or framework to test or discuss this. The conclusions are largely correlative.
Lack of assessment of research quality:
The review does not include a formal assessment of the risk of systematic error (e.g. using the RoB2 tool or the Newcastle-Ottawa Scale), which reduces the reliability of the synthesised conclusions, particularly in the case of interventions.
Variability of interventions and study groups:
The interventions used vary considerably from study to study (type of vitamin K, dosage, duration, route of administration), making it difficult to draw consistent conclusions. A more systematic comparative table of the most important study parameters would be useful.
Although most RCTs showed no effect of supplementation on renal function parameters, the authors do not attempt to critically analyse them (e.g. low doses, inadequate duration, population selection). A reference to the biological mechanisms underlying the lack of effects is needed.
Editor's comments:
There is a lack of consistency in citing the literature – some references are briefly described without giving methodological details (e.g. [17], [35]).
Some of the conclusions could be better structured (e.g. separate summaries of results for RCTs and observational studies).
Reviewer 2 Report
Comments and Suggestions for Authors
The current review evaluates the bidirectional relationship between vitamin K status and CKD, and whether vitamin K deficiency is a contributing factor or a result of CKD progression. The topic is timely and appropriate. I have the following questions:
- The central question ("cause or consequence?") is not appropriately addressed throughout the review. The review reflects both perspectives, but does not coherently or systematically present arguments for either.
- The mechanistic sections are underdeveloped. The review examines potential key mechanistic pathways, vitamin K–dependent activation of matrix Gla protein (MGP), oxidative stress, vitamin D or phosphate interdependence, but these pathways should be more explored and presented in a figure.
- Form/distribution (K1 vs K2) differences are described but there is no systematic comparison of pharmacokinetics, dietary sources or tissue-specific roles of vitamin K in CKD progression.
- The manuscript lacks consideration of confounders (i.e., phosphate binders, dialysis vintage, warfarin use) complicating vitamin K status in CKD patients.
Author Response
Please see the attached word document for response to your comments.

Round 2
Reviewer 2 Report
Comments and Suggestions for Authors
The paper can be accepted in its present form.